# Application of Immersed Membrane Bioreactor for Semi-Continuous Production of Polyhydroxyalkanoates from Organic Waste-Based Volatile Fatty Acids

**DOI:** 10.3390/membranes13060569

**Published:** 2023-05-31

**Authors:** Danh H. Vu, Amir Mahboubi, Andrew Root, Ivo Heinmaa, Mohammad J. Taherzadeh, Dan Åkesson

**Affiliations:** 1Swedish Centre for Resource Recovery, University of Borås, 501 90 Borås, Sweden; amir.mahboubi_soufiani@hb.se (A.M.); mohammad.taherzadeh@hb.se (M.J.T.); dan.akesson@hb.se (D.Å.); 2MagSol, Tuhkanummenkuja 2, 00970 Helsinki, Finland; magsol@kolumbus.fi; 3National Institute of Chemical Physics and Biophysics, 12618 Tallinn, Estonia; ivo.heinmaa@gmail.com

**Keywords:** biopolymer, immersed membrane reactor, polyhydroxyalkanoates, volatile fatty acids

## Abstract

Volatile fatty acids (VFAs) appear to be an economical carbon feedstock for the cost-effective production of polyhydroxyalkanoates (PHAs). The use of VFAs, however, could impose a drawback of substrate inhibition at high concentrations, resulting in low microbial PHA productivity in batch cultivations. In this regard, retaining high cell density using immersed membrane bioreactor (iMBR) in a (semi-) continuous process could enhance production yields. In this study, an iMBR with a flat-sheet membrane was applied for semi-continuous cultivation and recovery of *Cupriavidus necator* in a bench-scale bioreactor using VFAs as the sole carbon source. The cultivation was prolonged up to 128 h under an interval feed of 5 g/L VFAs at a dilution rate of 0.15 (d^−1^), yielding a maximum biomass and PHA production of 6.6 and 2.8 g/L, respectively. Potato liquor and apple pomace-based VFAs with a total concentration of 8.8 g/L were also successfully used in the iMBR, rendering the highest PHA content of 1.3 g/L after 128 h of cultivation. The PHAs obtained from both synthetic and real VFA effluents were affirmed to be poly(3-hydroxybutyrate-co-3-hydroxyvalerate) with a crystallinity degree of 23.8 and 9.6%, respectively. The application of iMBR could open an opportunity for semi-continuous production of PHA, increasing the feasibility of upscaling PHA production using waste-based VFAs.

## 1. Introduction

High production cost has been known as a critical drawback in the bioplastic commercialization of polyhydroxyalkanoates (PHAs). The production of PHAs mainly uses sugar or starch-based substrates as the main feedstock, which commonly accounts for a significant share of up to 30–50% in the overall production cost [1]. In order to increase the cost-competitiveness of PHAs, a variety of organic waste and residues derived from municipal and industrial processes have been evaluated to replace the expensive conventional substrate in PHA production [2]. These organic streams are abundant and rich in nutrients that can be used for the cultivation of PHA-storing bacteria. However, different types of pretreatments are required in order to access the fermentable compounds [3,4]. Additionally, recalcitrant and inhibitory materials could be released during the pretreatment processes, which potentially pose a number of negative impacts on PHA production [5]. In this regard, acidogenic fermentation is a promising approach to handling complex organic waste, producing fermentable compounds such as volatile fatty acids (VFAs) for the cultivation of PHA-bearing microorganisms [6,7,8].

VFAs have been known as important intermediates in anaerobic digestion for the production of biogas. However, due to the low profit of biogas, attention has been focused on VFAs, which can be used as precursors for further production of high-value biomaterials [6,8,9,10]. Recently, VFAs have emerged as economical feedstock for the production of PHAs [11]. Beyond their direct engagement in PHA biosynthesis, VFAs can be produced in large amounts from acidogenic fermentation of numerous organic waste streams, which can ensure a sufficient supply to replace expensive conventional carbon sources such as corn, wheat, sugar beet, etc. [12,13]. The use of VFAs in the cultivation of PHA-bearing bacteria has been conducted in several studies, showing promising results in PHA yields [14,15,16]. However, using VFAs, the PHA-bearing bacterial cultivation could be interfered with or hindered at high VFA concentrations of over 10 g/L [17]. This substrate inhibition, due to high VFAs content, may be the result of negative impacts on bacterial physiology, oxygen transport, osmosis, etc. [18]. Moreover, VFAs are mostly obtained as a mixture of different short-chain carboxylates, which could synergistically inhibit the fermentation process [19]. The aforementioned issues, therefore, have been responsible for the low PHA productivity in batch and fed-batch cultivations using high VFA concentrations [17,20].

In this regard, semi-continuous fermentation could be a solution in which high VFA concentration can be introduced to the culture at different time intervals to maintain favorable levels for bacterial growth via regulating the dilution rate. This cultivation method, however, can pose a disadvantage of bacterial washout from the removal of a fixed volume for the fresh feed to be replaced [21]. The loss of bacterial cells can greatly affect the system’s activity and output, as cell concentration is one of the major factors defining the bioconversion rate [22]. High bioconversion rates, due to high cell density, can improve bacterial tolerance to overcome the drawback of high substrate concentration. In order to achieve such conditions, immersed membrane reactors (iMBR), which have proven promising in cell retention, can be applied in semi-continuous cultivation modes to prevent the washout. In biotechnological processes, iMBR has been generally employed as a separation facilitator for the recovery of metabolites while retaining enzymes, bacteria, fungi, etc. [23,24,25]. The benefit of iMBRs in cell retention has been proven in a study by Wainaina et al. [26], where the bacterial concentration and its retention time were increased for the production of VFAs at high organic loading rates. The advantage of high cell concentration in substrate consumption at high concentrations was also observed in a study by Mahboubi et al. [27]. Following that concept, an iMBR can be utilized to overcome substrate inhibition during the cultivation of PHA producers at high VFA concentration.

In a novel attempt, this study aims to take advantage of iMBR to improve PHA productivity under high VFA concentrations under semi-continuous cultivation. To evaluate and compare the performance of iMBR, a semi-continuous cultivation without the assistance of a membrane was executed in parallel in a CSTR. The experiments using synthetic VFAs were initially conducted prior to the application of real VFA effluent. VFA consumption, cell density, biomass, and PHA production were determined to assess the efficiency of the MBR in preventing washout and increasing PHA productivity.

## 2. Materials and Methods

### 2.1. Microorganism and Preculture Preparation

*Cupriavidus necator* DSM 545 obtained from DSMZ-German collection of microorganisms and cell cultures GmbH (Leibniz Institute, Braunschweig, Germany) was loop-inoculated on nutrient agar plates composed of 1 g/L glucose, 15 g/L peptone, 6 g/L sodium chloride, 3 g/L yeast extract, and 15 g/L agar, incubated for 48 h at 30 °C, and then kept at 4 °C until use.

A minimal salt medium consisting of 10 g/L glucose, 2 g/L NaH_2_PO_4_, 1.8 g/L K_2_HPO_4_, 3 g/L (NH_4_)_2_SO_4_, 0.25 g/L MgSO_4_.7H_2_O, 0.01 g/L NaCl, 1 mL/L vitamins, and 1 mL/L trace elements was used for the precultures in shake flasks. Glucose, MgSO_4_, vitamins, and trace elements were autoclaved separately and mixed together with the main solution under sterile conditions with the addition of one loop inoculum. The medium was adjusted to pH 7 using NaOH 2M and incubated at 32 °C, 120 rpm for 40 h. Vitamin and trace metal solutions were prepared according to Vu et al. [28].

### 2.2. Organic Residues-Derived VFAs

In this study, the volatile fatty acid effluent was obtained from the acidogenic fermentation of apple pomace and potato peel liquor (PPL). A certain amount of VFAs was recovered every day using a microfiltration process, which was applied to support the separation of VFA solution and undigested substrate. The filtrate obtained was preserved at −20 °C before being thawed in a cold room at 5 °C for use as substrate in PHA production. The composition of the final VFA solution is described in Table 1.

### 2.3. Immersed Membrane Bioreactor Setup

The flat sheet membrane applied in this study was the 2nd generation Integrated Permeate Channel (IPC) back-washable membrane panel obtained from Flemish Institute for Technological Research (VITO NV, Mol, Belgium). The membrane was coated by hydrophilic polyethersulfone (PES) possessing inbuilt air diffusers (at the bottom of the membrane) with a pore size of 0.3 µm to create better aeration for fouling prevention, facilitating the filtration process. With an effective area of 68.6 cm^2^, the panel provided a permeability of 3000–4000 L/h·m^2^·bar. The membrane was integrated inside a 2.5 L bioreactor (Biostat B plus, Sartoius BBI Systems GmbH, Melsungen, Germany) to prevent washout during the filtration using a peristaltic pump (Watson Marlow 400, Wilmington, MA, USA) before a daily feeding regime. The filtration was conducted with the backwashing-assisted protocol, the process of which was set with a 3.5 min cycle (30 s of backwash and 3 min of permeation) and controlled by a Schneider Zelio logic relay (Schneider Electric Automation GmbH, Lahr, Germany). The permeate flow rate and transmembrane pressure (TMP) were measured by an Atrato 710-V11-D ultrasonic flowmeter (Titan Enterprises Ltd., Sherborne, UK) and microfluidic pressure sensor (KELLER, Basel, Switzerland), respectively. The recorded raw data were transferred to a computer connected to measurement instruments.

Before the experiment, the membrane was chemically cleaned in order of 2% NaOH, 1% H_3_PO_4_, and 200 ppm NaClO at 40 °C for 45 min. Sterile distilled water was used to rinse the membrane, which afterward was connected to autoclaved reactor.

### 2.4. Semi-Continuous Cultivations

Regarding semi-continuous cultivation, the same minimal salt medium described in Section 2.1 was used in both reactors with and without membranes. Glucose was subsequently replaced by a mixture of synthetic VFA and real VFA effluent. Nutrient solution MgSO_4_ and VFA medium were autoclaved separately and mixed together with the addition of 100 mL seeding cultures (Section 2.1) before starting the experiment. During the cultivations, the reactors were maintained at 32 °C, pH 7, 150 rpm, and with aeration of 1 vvm. The parameters were monitored and controlled by a Biostat B plus fermentation controlling unit. Fatty acid ester antifoam was manually added in case of foaming.

After acclimatization of 32 h, a daily feeding regime of 300 mL fresh substrate was conducted, followed by the removal of the same amount of solution to leave a working volume of 2 L. Regarding the reactor without the membrane, 300 mL solution was directly collected from the sampling tube. In iMBR, this amount was only 50 mL, while the rest was recovered by filtration to prevent washout. Samples taken from sampling tubes were used for the analysis of substrate consumption, cell density, biomass production, and PHA accumulation.

With a total concentration of 8.81 g/L, the real VFAs could not cope with the dilution ratio of 6.6 to maintain the concentration of 5 g/L in every feeding. Therefore, the feeding regime was modified to twice per day after 16 and 8 h from the first 32 h of cultivation. In this regard, the nutrients could be available throughout the cultivation with a total VFA provision of 2.5 g/L per day.

### 2.5. Analytical Methods

The changes in VFA consumption during the fermentation were measured using high-performance liquid chromatography (HPLC) (Waters 2695, Waters Corporation, Milford, CT, USA). The HPLC used a hydrogen-based column equipped with an ultraviolet absorption detector at 210 nm wavelength for VFA quantification (Waters 2487, Waters Corporation, Milford, CT, USA). The analysis was conducted at 60 °C using a mobile phase of 5 mM H_2_SO_4_ at a rate of 0.6 mL/min.

The changes in cell density were analyzed by the measurement of optical density in the mode of linear range at 600 nm using a Libra S60 (Biochrom Ltd., Cambridge, UK) spectrophotometer.

Dry cell weight (DCW) measurement was conducted from 5 mL of culture taken at specific intervals. The bacterial cells were harvested by centrifuging at 9000× *g* for 5 min (Megafuge 8, Thermo Fisher Scientific GmbH, Dreieich, Germany), followed by a repeating rinse with Milli-Q water. The biomass was transferred to a pre-weighted aluminum cup for a drying process of 24 h at 70 °C until a constant weight was reached.

For PHA extraction, a rinsed biomass obtained from centrifugation of 25 mL solution (at 9000× *g*, 5 min) was incubated in 10 mL of 2% NaClO at 70 °C for 1 h [28]. After incubation, the solution was centrifuged (9000× *g*, 5 min) and rinsed with Milli-Q water three times before the extracted pellet was dried in a pre-weighted aluminum cup at 70 °C for 24 h.

### 2.6. PHA Characterization

Fourier Transform Infrared (FTIR) spectroscopy (Nicolet iS10, Thermo Fisher Scientific, Waltham, MA, USA) was used for the analysis of functional groups from extracted samples. A scan conducted 32 times in a spectrum of 400 to 4000 cm^−1^ using Nicolet OMNIC 4.1 software. The obtained spectrum were analyzed using Essential FTIR software 3.5 (eFTIR, Madison, WI, USA) for the final results. 

Thermal stability of the extracted PHAs as analyzed using thermogravimetric analysis (TGA) (Q500 TA instruments, Waters LLC, New Castle, DE, USA). Approximately 5 mg of samples were heated from room temperature to 700 °C at a range of 20 °C/min under a nitrogen atmosphere. The analysis was conducted in triplicates.

Thermal properties of extracted PHAs were determined by differential scanning calorimeter (DSC) analysis (QA500 TA instruments, Waters LLC, New Castle, DE, USA). Approximately 5 mg of samples enclosed by an aluminum pan were heated from −40 °C to 225 °C at a rate of 10 °C/min under a nitrogen atmosphere. 

^13^C MAS NMR spectra were obtained by a Bruker AVANCE-II spectrometer using a magnetic field strength of 14.1 T at a frequency of 150.9 MHz. A custom-made MAS NMR probe was used for 25 × 4 mm Si_3_N_4_ rotors, and the sample spinning frequency was set to 12.5 kHz. In this experiment, the relaxation delay was 5 s. Cross polarization (CP) was used to record the NMR spectra using a ramped spin-locking pulse in the ^1^H channel together with a pulse in the ^13^C channel of suitable amplitude. In the experiment, using a single pulse on the ^13^C channel with ^1^H decoupling, a 90° pulse (3µsec) and a 60 s relaxation delay was applied.

## 3. Results and Discussion

The current study aimed to conduct PHA production in a semi-continuous fermentation mode to tackle the low productivity of PHA observed in batch conditions. In this regard, a membrane bioreactor and a CSTR were compared in performance when fed VFAs. 

### 3.1. Semi-Continuous Cultivation of C. necator Cultivation Using Synthetic VFAs in CSTR

Compared to other valuable secondary metabolites (e.g., antibiotics, enzymes) that are secreted during bacterial fermentation, PHAs are intracellular compounds that can only be recovered together with the biomass. Therefore, PHA production has been commonly conducted in batch cultivation on a large scale [29]. Regardless of the simplicity, batch cultivation faces the critical drawback of low productivity in PHA yield [30]. In this study, PHA production was conducted in a continuous mode using similar VFA content and cultivation factors that were thoroughly optimized in batch fermentation in a previous study by Vu et al. [31]. The changes in total VFA and cell density corresponding to biomass production and PHA accumulation are presented in Figure 1.

As presented in Figure 1c, semi-continuous fermentation provided a maximum biomass of 4.5 g/L, corresponding to the highest PHA concentration of 2.0 g/L after 80 h. Compared to the results achieved in batch cultivation by Vu et al. [31], the obtained results were 1.6 and 1.3 times higher in terms of biomass and PHA production, respectively. However, the production yield gradually declined as the cultivation proceeded due to the reduction in cell density (Figure 1b). In fact, after the first feeding at 32 h, a certain number of bacterial cells have already been washed out of the bioreactor during medium recovery and the addition of fresh substrate (every 24 h). This washout occurred as the dilution rate was beyond the maximum specific cell growth rate according to the Monod growth kinetics of *C. necator* [32]. The loss of bacterial cells, in this case, negatively affected the cultivation, presented in the form of slow VFA consumption, leading to the steady pileup of VFA after each feeding cycle.

The accumulation of VFAs can also be attributed to the selectivity in substrate consumption, which is considered a detrimental factor in the current semi-continuous fermentation. Similar to the findings by Yun et al. [17] and Setiadi et al. [33], butyrate was found to be the preferred substrate, which was previously consumed among other VFAs. Acetic acid, on the other hand, was the least favorable carbon source (Figure 1a). The consumption of acetic acid was rather marginal, while other VFAs were almost depleted. The same pattern of VFA utilization was recorded after every feeding cycle. According to Figure 1b, the system could not endure fermentation conditions after 80 h, where 13% of the bacterial cell had been removed, and the total VFA concentration reached almost 10 g/L. At high concentrations, VFAs can induce inhibition in bacterial growth due to changes in bacterial physiology [18]. This hypothesis is in line with the studies by Yun et al. [17] and Agustín Martinez et al. [20], where the cultivation was delayed or completely inhibited as the VFAs’ concentrations were above 10 g/L. Therefore, in this study, the loss in bacterial cells, together with the accretion of acetic acid, were considered the main reasons leading to the failure of the system.

### 3.2. Semi-Continuous Cultivation of C. necator Using Synthetic VFAs in iMBR

It was observed that in the CSTR cell washout was a critical factor causing the accumulation of VFA and hindering continuous fermentation. Therefore, an immersed membrane bioreactor (iMBR) was applied as a novel approach to prevent washout, facilitating the fermentation in semi-continuous mode to achieve a higher yield. Membrane separation has been widely applied for the separation and recovery of metabolites in biotechnological processes [23,24]. In the current study, an iMBR was used for cell retention during the filtration of a free-nutrient medium for the provision of the fresh substrate, extending the cultivation time.

The efficiency of the MBR in cell retention can be observed in Figure 2b, in which the cell concentration was well maintained after three cycles of feeding at the VFA concentration of up to 33 g/L. Compared to CSTR, the bacterial cells in iMBR can keep increasing over time, overcoming the significant washout at 80 h. The uptrend remained afterward at a moderate rate, leading to a maximum cell density of 26.2 at the end of the cultivation. A marginal decrease in bacterial concentration was only recorded at 104 h, which can be ascribed to the effect of membrane pore size and the type of bacteria. As reported in a study by Gaveau et al. [34], Gram-negative bacteria were found to be able to deform, passing through the microfiltration membrane due to the flexibility of the peptidoglycan cell wall. *C. necator* is a Gram-negative bacteria with an average diameter of 0.4–0.6 µm, which may partly penetrate through the applied membrane with an average pore size of 0.3 µm. Overall, the filtration process was conducted in a fluctuating flux within an average range of 76–88 L per m^2^/h (Figure 3a). Under this condition, the membrane performed competently to handle a biomass concentration of 1.6 g/L in the first cycle, presenting a gradual increase in TMP of up to 0.45 bar. TMP values, however, were increased significantly to an extreme value of 0.9 bar in the next cycles, followed by an increase in bacterial cells from 3.4–6.5 g/L. The biomass amount till this point was similar to the TSS values in the first 32 days of VFA production and recovery by Pervez et al. [35]. However, the filtration process of which was operated in a much lower flux of 10.5 L per m^2^/h, resulting in a stable and low TMP of 6–10 mbar. Therefore, the high TMP in this study can be attributed to the current flux, which was much higher than that of 10–20 L per m^2^/h applied on an industrial scale [36]. The high flux, moreover, can increase issues associated with the filtration process, such as pore clogging and cake layer formation, by promoting the deposition and attachment of bacteria on the membrane surface and inside membrane pores at high cell concentrations [37]. Furthermore, regardless of the physical membrane cleaning with backwashing, the filtration time was extended over time along with an increased TMP (Figure 3b). 

As the majority of bacterial cells were retained in the bioreactor, the cell concentration can keep increasing over time, enhancing the bioconversion of VFAs. Regardless of the same pattern of VFA consumption in the first 32 h, as in CSTR, VFA assimilation in MBR was accelerated, leading to the elimination of acetic acid accumulation as this component was simultaneously consumed during the fermentation (Figure 2a). The performance of the system, therefore, was significantly improved with an obtained maximum biomass of 6.6 g/L (Figure 2c), corresponding to the highest PHA content of 2.8 g/L. The results obtained were 1.5 and 1.4 times higher than that of the CSTR, respectively. The benefit of high cell density was also observed in a study by Khomlaem et al. [38] using ceramic filtration, in which the biomass and PHA production were increased 10 times compared to that of in-batch cultivation using a corn cob hydrolysate. A similar concept can be found in a study by Aloui et al. [39], which utilized a membrane reactor to achieve feasible PHA production using a high concentration of glucose, up to 40 g/L. 

Regardless of a steady increase in both biomass and PHA production, the PHA yield on biomass in this study decreased over time from 50% to 40% at the end of the cultivation. The drop in PHA yield can be attributed to the similar C/N ratio of 6 applied in this study which was found to be optimal for biomass production [31]. VFAs were almost depleted in the MBR at the end of every 24 h cycle. Therefore, the C/N ratio was presumably maintained in every subsequent feeding, continuously encouraging biomass synthesis and extending the gap with the PHA accumulation. The maximum PHA achieved of 2.8 g/L, nevertheless, was almost double the amount obtained in other studies by Chakraborty et al. [40] and Yun et al. [17], (in a range of 1–1.6 g/L).

In contrast to growth-associated PHA-accumulating bacteria, such as *Azohydromonas lata*, *C. necator* belongs to a bacterial group that requires restricted conditions to accumulate PHAs [41]. Therefore, a two-stage cultivation has been developed for non-growth-associated bacteria to obtain sufficient productivity. In the first stage, the bacteria are supplied with a balanced C/N ratio medium to optimize biomass production. The rich-biomass solution is afterward transferred to a second reactor provided with an excess carbon source for the second phase of PHA accumulation. In an MBR, two-stage cultivation can be conducted in one single reactor. Exhausted medium in the first stage can be removed by a filtration process, while the bacterial cells are retained for the next stage with the addition of fresh medium. This, in turn, intensifies PHA production, minimizing the risk of contamination and increasing cost competitiveness.

### 3.3. C. necator Cultivation Using Organic Residues-Derived VFAs in iMBR

Prior to starting cultivation, organic residues-derived VFA effluent was originally diluted to obtain a similar concentration to that used in synthetic VFAs of 5 g/L. Compared to synthetic VFAs, the concentration of butyric acid in a real VFA stream was higher. The deviation of which, however, was expected to not significantly affect cultivation as butyric acid was proven to be a preferable carbon source for *C. necator* [31,33]. This can be observed in Figure 4a, where the trend in VFA consumption remained with the fast uptake of butyric acid and slow assimilation of acetic acid. In contrast, the major differences were derived from the lower concentration of real VFA effluent (8.81 g/L). Regardless of the change in the feeding regime to maintain VFA availability, the concentration of VFAs after each feeding cycle was 44–76% lower than that of CSTR and MBR running on synthetic VFAs. This, in turn, resulted in faster VFA depletion throughout the fermentation using real VFA effluent (Figure 4b). In MBR, the overall productivity of the system can be potentially increased at higher substrate concentrations [26]. However, the lower concentration in organic residues-based VFAs effluent, in this case, could not ensure the required concentration in the following feeding stream to support bacterial growth after 80 h. Therefore, the maximum biomass concentration was around 3.3 g/L, which was lower than that in CSTR and MBR, with differences of 26 and 50%, respectively. The decrease in biomass obtained consequently lowered the maximum PHA yield (1.8 g/L). The accumulated PHA, furthermore, was reduced afterward, while the bacterial cells remained, leading to the gradual decline in PHA yield on biomass (Figure 4c). The decrease in PHA content could be attributed to the internal consumption of PHA, which was triggered for cell maintenance as bacterial growth was halted [42]. 

In the aspect of filtration performance, lower biomass concentration was partly beneficial for the membrane operation. In particular, the increase in cell concentration can result in an increase in viscosity and the deposition of bacterial cells on the membrane surface, which can gradually raise the TMP [27]. This behavior can be observed in Figure 5b, in which the TMP in the first two days rose moderately, corresponding to the increase in the biomass concentration from 1.3 to 2.5 g/L (Figure 4c). Similar to the filtration in MBR using synthetic VFAs, the TMP in this experiment only reached the extreme level of 0.9 bar as the biomass content was around 3.3 g/L. Lower cell density, moreover, could mitigate the process of membrane fouling under a high permeate flux of 78–92 L per m^2^/h, thereby reducing the filtration time (Figure 5a). 

### 3.4. PHA Characterization

#### 3.4.1. Fourier-Transform Infrared Spectroscopy (FTIR) Analysis

The FTIR spectra of all extracted samples were compared with the commercial poly(β-hydroxybutyrate-β-hydroxyvalerate) PHBV in Figure 6. It can be observed that all spectra obtained did resemble commercial PHBV, with typical peaks of 1719 cm^−1^ and 1379 cm^−1^, correlating to the stretching in ester carbonyl (C=O) group and vibration of –CH group, respectively. At the same time, the bands at 2900 to 2800 cm^−1^ demonstrate the presence of methyl (CH_3_) and methylene (CH_2_) groups. The transmittance band from 1300 –1000 cm^−1^ represented characteristic peaks of C=O bonding in ester group and C-O-C functional groups, which are representative of PHA functional groups [43,44]. Therefore, it can be concluded that the PHA obtained in this study was a copolymer of PHBV, the FTIR spectra of which were also similar to PHBV produced in other studies using different carbon sources [45,46,47]. 

#### 3.4.2. Thermogravimetric and Differential Scanning Calorimetry Analysis (TGA and DSC)

Overall, the thermal degradation of extracted PHA samples was similar, sharing the same range in both initial decomposition temperature (T_onset_) and maximum degradation temperature (T_max_) (Table 2). A slight difference was observed in the total mass loss, which is normally 10% lower in samples obtained from the VFA stream due to the excess impurity content [31]. From the aspect of thermal properties, regardless of the analogous melting temperatures (T_m_), the degree of crystallinity of the polymer extracted from the VFA stream was only 9.6%. This value was significantly lower than that of synthetic VFAs due to the small fraction of impurities derived from recovered organic residues-based VFAs. The results obtained, however, were found to be the same compared to PHA samples from studies by Nygaard et al. [48] and Ntaikou et al. [49]. According to Rosengart et al. [50], the low degree of crystallinity could be an advantage for widening the processing windows of PHAs.

#### 3.4.3. Nuclear Magnetic Resonance (NMR) Analysis

The 13C CPMAS and the direct excitation NMR spectra were measured from both samples. Figure 7 shows a comparison of the CPMAS NMR spectra.

There do not appear to be large differences in the NMR spectra. However, the 3HB methyl peak area contains two overlapping peaks, one more narrow and one broader. These are typically assigned to the crystalline and amorphous regions of the 3HB, respectively [51]. The narrow peak to higher frequency is assigned to methyl groups in more ordered chains of the HB in the trans arrangement of the backbone polymer, while the broader peak is assigned to methyls in a mixture of trans gauche arrangements of chains in the more disordered regions. This disordered, or amorphous, region has considerably more motion than the ordered regions.

When comparing the two samples, some small differences can be seen within this methyl area. This area is shown in Figure 8 for both the CPMAS and direct excitation NMR experiments. The CPMAS NMR spectra are not exactly the same as the direct excitation NMR spectra, as shown in Figure 8, but the same general phenomenon can be seen in both, i.e., there appears to be a slightly higher peak due to the more ordered polymer chains in the synthetic VFA sample.

The ^13^C T_1_ for the methyl carbons of the narrow, ordered component of 3HB has been measured as 2.8 s at 270 MHz (for ^1^H), while that of the broad component was 0.3 s [52]. In the direct excitation MAS NMR experiments here recorded at 600 MHz (for ^1^H), a 60-s relaxation delay was used. It is likely that the T_1_ at this field is not more than 12–15 s, so this relaxation delay should be enough for reasonably quantitative data from the methyl groups in the direct excitation NMR spectra shown in Figure 8.

In order to try and quantify the degree and nature of the crystallinity of the two samples, both the 3HB methyl area and the HV_4_ side chain area from the direct excitation NMR experiments were compared. This comparison is shown in Figure 9.

For the HB, there is not much difference in the number of methyl groups in the ordered regions in both samples, even though the DSC results (Table 2) show significantly higher crystallinity in the synthetic sample. In addition, the overall crystallinities measured by DSC are much lower than the amount of ordered HB found from the HB methyl peaks in Figure 9. The main difference seen in the NMR spectra is that the linewidth of the ordered component is somewhat smaller in the synthetic VFA compared to the real VFA sample. Since solid-state NMR measures local order on a much smaller scale than DSC, it is assumed that the difference in the crystallinities by NMR and DSC is due to the fact that in these samples, the local order of the trans chains arrangements occurs in very small areas that are not big enough to give a significant melting exo/endotherm in DSC. It is also possible that NMR measures order even from chains locally trans-ordered within the amorphous or interfacial region [53]. 

There are also differences in crystallinity between the two samples by DSC, with the synthetic VFA having a higher crystallinity than the real VFA sample. NMR shows about the same degree of order for both samples (around 60%). The ordered methyl peak from the synthetic VFA around 21 ppm, however, gives a narrower peak. In the peak deconvolutions shown in Figure 9, the synthetic VFA has a linewidth of 88 Hz, while the real VFA sample has 98 Hz. This could indicate that the ordered domain areas within the synthetic VFA are larger, leading to smaller variations in the trans chain arrangements. Larger ordered domain areas could give more areas detectable as crystalline by DSC and more narrow NMR linewidths.

It is interesting to note that the HV_4_ side chain CH_2_ peaks around 30 ppm; previously assigned to some small amounts of pure amorphous HV homopolymer in the copolymer [31], appears to be slightly more intense in the synthetic VFA. Since the other peaks in this area, 25–28 ppm, are due to the HV side chain CH_2_ in the 3HB/HV copolymer, this would imply that there is less HV within the copolymer structure than in the synthetic VFA. If these act as defect sites, then the lack of some HV in the 3HB/HV part of the synthetic VFA could enhance the ability of the 3HB to form larger ordered regions. 

The comparison of DSC crystallinity and NMR order is interesting since it indicates that there is a lot of order within the samples, 60% by NMR, but that the ordered domains are mostly quite small, as shown by the much smaller degree of crystallinity by DSC (9.6 to 23.8%).

## 4. Conclusions

Substrate inhibition and accumulation of acetic acid were found to be major hindrances in the cultivation of *C. necator* DSM 545 using a high concentration of VFAs. In this study, iMBR-assisted fermentation using a semi-continuous mode was able to increase bacterial tolerance by maintaining the cell concentration and manipulating the VFA concentration. In an experiment using synthetic VFAs, the system could operate at a maximum feeding concentration of 33 g/L, yielding the highest PHA production of 2.8 g/L after 128 h cultivation. The same concept was applied for cultivation using an organic waste-based VFA effluent with a total concentration of 8.8 g/L to produce a maximum PHA content of 1.8 g/L. The PHA obtained was found to be a copolymer of PHBV with a degree of crystallinity of 9.6%. The application of iMBR can be further applied in two-stage production of PHA, in which both of them can be conducted in the same reactor, lowering the footprint in the processing stages.

## Figures and Tables

**Figure 1 membranes-13-00569-f001:**
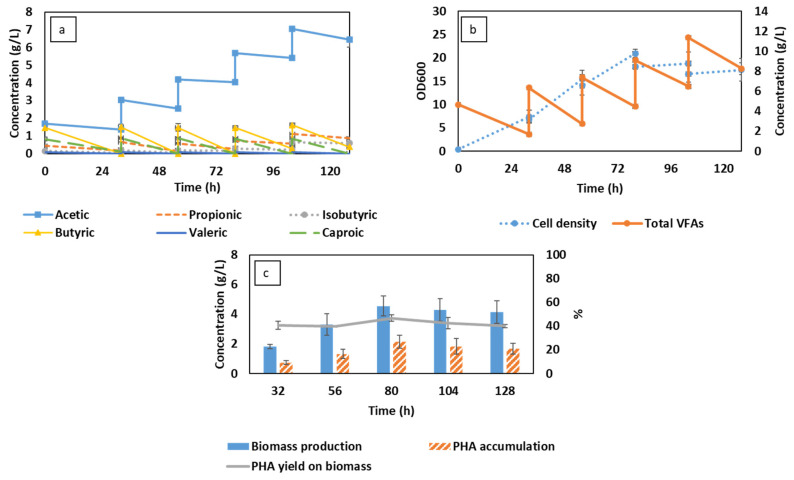
Changes in volatile fatty acid (VFA) compositions (**a**), total VFA concentration, and cell density (**b**) corresponding to biomass production and PHA accumulation (**c**) during the semi-continuous fermentation using synthetic VFAs.

**Figure 2 membranes-13-00569-f002:**
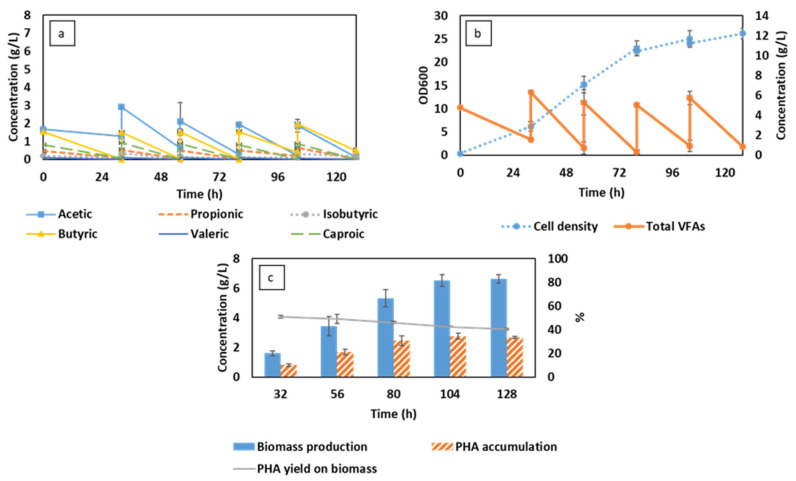
Changes in volatile fatty acid (VFA) compositions (**a**), total VFA concentration, and cell density (**b**) corresponding to biomass production and PHA accumulation (**c**) during MBR-integrated semi-continuous fermentation using synthetic VFAs.

**Figure 3 membranes-13-00569-f003:**
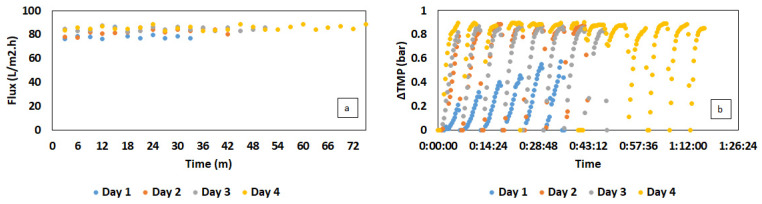
Changes in transmembrane pressure (TMP) (**a**) and permeate flux (**b**) during each cycle of filtration in MBR-integrated continuous fermentation using synthetic VFAs. The flux data were presented in the form of an average value for 3 min of filtration.

**Figure 4 membranes-13-00569-f004:**
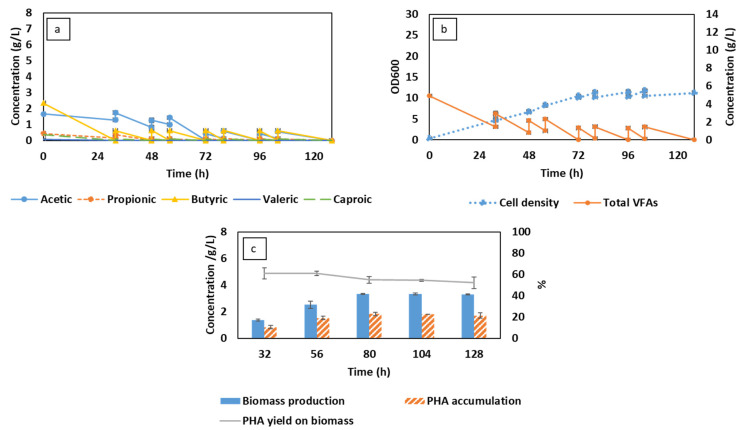
Changes in volatile fatty acid (VFA) compositions (**a**), total VFA concentration, and cell density (**b**) corresponding to biomass production and PHA accumulation (**c**) during MBR-integrated semi-continuous fermentation using organic residues-derived VFAs.

**Figure 5 membranes-13-00569-f005:**
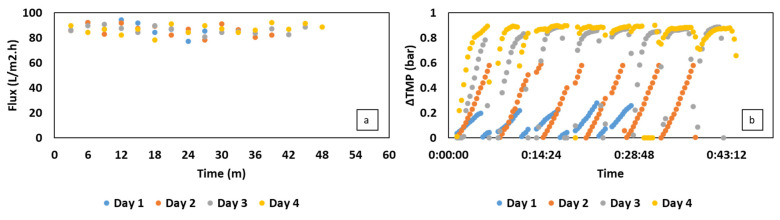
Changes in average transmembrane pressure (TMP) (**a**) and permeate flux (**b**) during two cycles of filtration in MBR-integrated continuous fermentation using organic residues-based VFAs. The flux data were presented in the form of an average value after a 3-min filtration.

**Figure 6 membranes-13-00569-f006:**
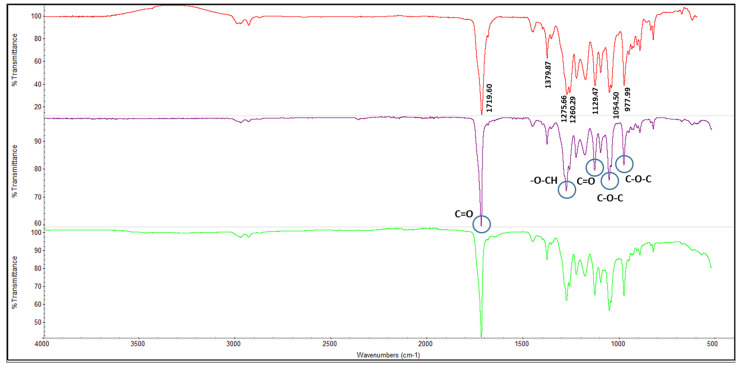
Comparison of Fourier-Transform Infrared Spectriscopy (FTIR) spectra between commercial poly(β-hydroxybutyrate-β-hydroxyvalerate) (PHBV) (red line) and extracted polyhydroxyalkanoate (PHA) obtained from cultivation of *C. necator* DSM 545 in a different medium of synthetic VFAs (purple line), and real VFA effluent (green line).

**Figure 7 membranes-13-00569-f007:**
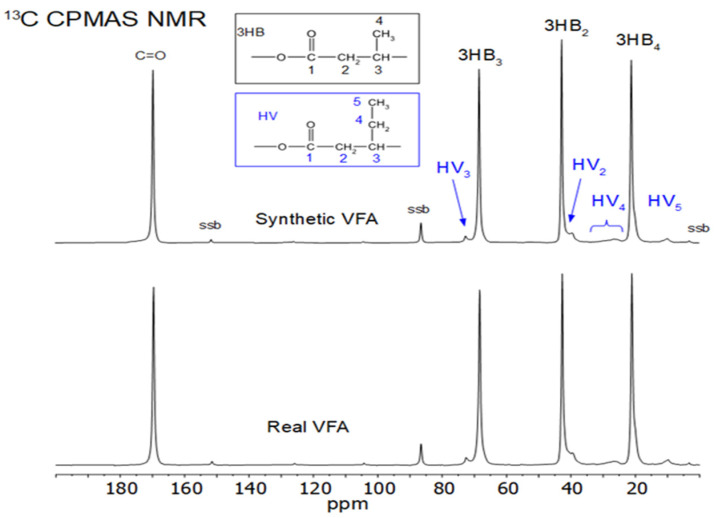
^13^C CPMAS NMR spectra of Synthetic VFAs and Real VFAs.

**Figure 8 membranes-13-00569-f008:**
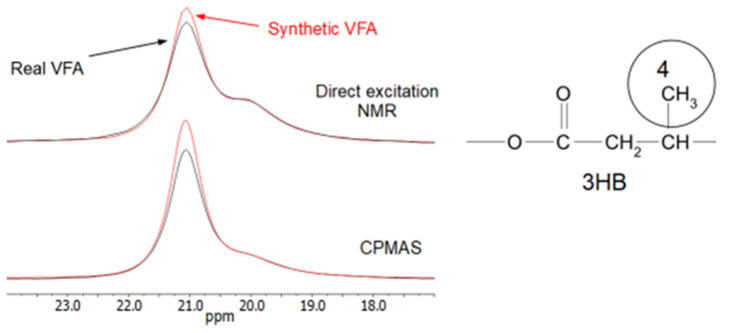
Comparison of the HB methyl area of the CPMAS and direct excitation NMR spectra of Synthetic VFAs and Real VFAs.

**Figure 9 membranes-13-00569-f009:**
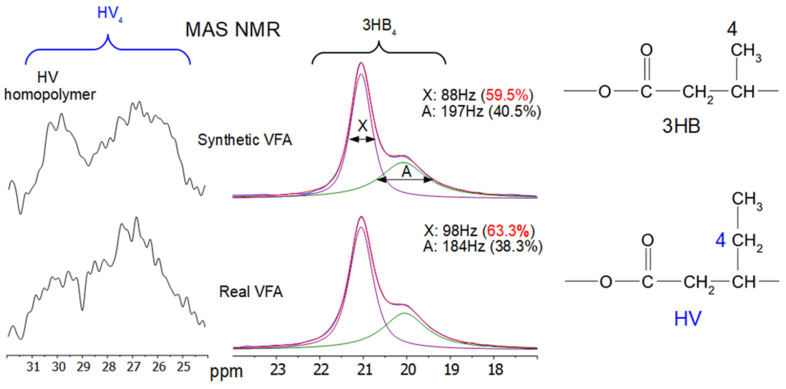
Comparison of the 3HB_4_ and HV_4_ areas of the direct excitation MAS NMR spectra from synthetic VFAs and real VFAs. The peak labelled X is due to the methyl carbons of the 3HB polymer associated with the ordered areas while the peak labelled A is associated with the 3HB polymer methyl carbons in the more disordered areas. The linewidths and populations (in brackets) of each peak are as shown for each sample.

**Table 1 membranes-13-00569-t001:** The composition of the final volatile fatty acid (VFA) solution obtained from the acidogenic fermentation of apple pomace and potato peel.

Components	Concentration (g/L)
Acetic acid	3.01
Propionic acid	0.80
Butyric acid	4.15
Isobutyric acid	0.07
Valeric acid	0.13
Isovaleric acid	0.00
Caproic acid	0.65
Ammonium	0.84
Total VFAs	8.81

**Table 2 membranes-13-00569-t002:** Comparison of thermal properties of extracted polyhydroxyalkanoate (PHA) obtained from the cultivation of *C. necator* DSM 545 in different mediums.

Samples	Tonset (°C)	Tmax (°C)	Total Mass Loss (%)	Tm (°C)	Melting Enthalpy (J/g)	Degree of Crystallinity (%)
Synthetic VFAs	246.8.6 ± 1.8	297.6 ± 2.0	98.1	146.2 ± 0.5	34.5 ± 1.3	23.8
Organic waste-based VFA effluent	253.16 ± 1.2	302.3 ± 0.8	89.1	143.4 ± 0.9	13.8 ± 0.8	9.6

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
