# Peer review of "Application of Immersed Membrane Bioreactor for Semi-Continuous Production of Polyhydroxyalkanoates from Organic Waste-Based Volatile Fatty Acids"

_membranes, 2023, doi:10.3390/membranes13060569_

Round 1

Reviewer 1 Report

Comments and Suggestions for Authors

I read carefully an interesting and comprehensive research article entitled Application of immersed membrane bioreactor for semi-continuous production of polyhydroxyalkanoates from volatile fatty acids. The concept of the article is interesting and suitable for ‘Membranes’ This manuscript is generally well written and clearly presented however still needs to address some comments, and thus require moderate revision to improve the quality of the manuscript.

1.     Title should modify which can describe the whole research work need to mention the source of VFAs.

2.     In the introduction discuss briefly recent advances related to VFAs utilization for various value-added products Polymers 13 (24), 4297, 2021 is required.

3.     Foe Line no 37-38 elaborate the different wastes used for PHA production and their limitations Industrial crops and products 150, 112425, 2020.

4.     Compare your results with other studies by adding one table for the same.

5.     Have authors checked the advantage of utilization VFAs with pure or waste derived carbon source for PHA production?

6.     Techno Economic challenges and limitations of this system should be included? Add future research directions also.

7.     The conclusion of the study needs to be added with the specific output obtained from the study, it could be modified with precise outcomes with a take home message.

8.     Some English and grammar mistakes are present that need to be correct to improve the quality of the manuscript.

Comments on the Quality of English Language

 Some English and grammar mistakes are present that need to be correct to improve the quality of the manuscript. Authors need to check during revision stage.

Author Response

We thank the reviewer for the comment. Please find our response in the attached word file.

Reviewer 2 Report

Comments and Suggestions for Authors

The manuscript entitled “Application of immersed membrane bioreactor for semi-continuous production of polyhydroxyalkanoates from volatile fatty 3 acids” presents the results from the use of an immersed membrane bioreactor (MBR) for the production of PHAs using synthetic and real VFA solution and the Cupriavidus necator microorganism. The authors also assess the physicochemical characteristics of the produced PHAs in terms of its crystallinity and other quality parameters. The work to the reviewer’s best knowledge is original and the experimental procedure is well organized. The authors operate a CSTR and a MBR reactor assessing the differences on their performance. It is also interesting that they provide detailed description on the quality properties of the PHAs produced, depending on the VFA feed characteristics. The results are basically presented in two groups: the results that refer to the bioreactors performance assessment and the results concerning the properties of the produced PHAs. While, the second part is very well-written and presented in detail, the first part has some flaws both concerning the quality of the manuscript’s  text, as well as some minor flaws concerning the presentation of the results and their interpretation. Moreover, the authors are kindly asked to provide some more detail concerning the current state-of-the-art for PHAs production employing MBR technology. More specifically:

1.     Although the authors do present some data concerning the current trends on PHAs production, they fail to provide details on already presented data concerning PHAs production in MBRs. Indicatively, the following studies report results from PHAs production in MBRs, however, they are not critically assessed by the authors: DOI:10.1016/j.cej.2022.138641; DOI:10.3390/membranes12070703; DOI:10.1016/j.ijbiomac.2022.05.089; DOI:10.1016/j.ijbiomac.2021.09.180; DOI: 10.1016/j.bej.2020.107687

2.     The quality of the English language is moderate in some parts of the manuscript. The authors are kindly asked to carefully edit the syntax of the following lines to render them more clear to the reader: Line 15, Lines 20-21, Lines 22-23, Line 25, Line 39, Line 51-52, Line 54, Line 62, Line 69-70, Line 71 (… by [21]), Line 110, Line 112, Line 125, Lines 144-146, Lines 241-242, Lines 258-259, Line 264,  Lines 283-284, Lines 286-288, Figure 1 and 3 legends, Line 311, Lines 313-314, Line 316, Lines 329-330, Line 450, Lines 453-455.

3.     Line 124: The authors mention “Eq. (1) and (2)” without providing these equations.

4.     Lines 130-131: The authors do not mention the concentration of the carbon substrates used in this study.

5.     Line 138: Can the authors please explain why they decided to remove the fermented broth after (and not before) adding the new substrate (i.e. VFA solution).

6.     Line 165: Can the authors provide some reference for the analytical methods that they employ?

7.     Line 246: Figure 2b does not show results concerning the cell rejection efficiency. It presents the temporal evolution of the filtration flux…

8.     Lines 281-283: The authors mention that the “productivity of the system, therefore, was significantly improved with the obtained maximum biomass of 6.6 g/L (Figure 3c), 1.5-times higher than that of the CSTR.” However, the higher final cell concentration (fermentation titer) doesn’t necessarily mean that the total productivity is higher since during the CSTR operation 1.2L of fermentation broth containing cells were removed from the bioreactor during the 120-hours fermentation. On the contrary, during the MBR operation the volume of the removed broth (for sampling) was only 0.2L. Therefore, the authors are kindly asked to calculate the total cell biomass produced during the fermentation (including the removed quantities) and report these data before commenting on the overall processes productivities.

9.     Lines 338-339: The increase of the viscosity of the fermentation broth is not the main cause of membrane fouling. The authors should take into consideration the increase of the deposition rate of rejected cells with the increase of the cells’ concentration and the “critical flux” concept in relation to cells/biomass concentration.

10.  The authors fail to mention any conclusions (in the conclusions section) concerning the 2nd part of their study on PHAs properties in relation to feed VFAs.

Finally, the reviewer has a comment concerning the use of the term semi-continuous in the manuscript. To the reviewer’s best knowledge, a process is considered “semi-continuous” when both the feed (reactants) and the final product are removed from the reactor for discrete time points. In this manuscript, the bioreactor operation without the membrane is semi-continuous since the product (PHA) is removed during each feeding. However, the MBR operation is semi-batch, since no product removal takes place (apart from the sampling quantity) and the product is finally recovered at the end of the process. So, the MBR operates in semi-batch mode, i.e. the feed is gradually added during the process at discrete time points, while the product is recovered at the end of the process.

Comments on the Quality of English Language

The quality of the English language is moderate in some parts of the manuscript. The authors are kindly asked to carefully edit the syntax of the following lines to render them more clear to the reader: Line 15, Lines 20-21, Lines 22-23, Line 25, Line 39, Line 51-52, Line 54, Line 62, Line 69-70, Line 71 (… by [21]), Line 110, Line 112, Line 125, Lines 144-146, Lines 241-242, Lines 258-259, Line 264,  Lines 283-284, Lines 286-288, Figure 1 and 3 legends, Line 311, Lines 313-314, Line 316, Lines 329-330, Line 450, Lines 453-455.

Author Response

We thank the reviewer for the careful consideration of the manuscript. Please find our response in the attached word file. 

Reviewer 3 Report

Comments and Suggestions for Authors

Comments on the Quality of English Language

Authors should check the typos and grammatical errors.

Author Response

We thank the reviewer for the comments. Please find our response in the attached word file. 
